# Characterization of Quality Properties in Spoiled Mianning Ham

**DOI:** 10.3390/foods11121713

**Published:** 2022-06-11

**Authors:** Yanli Zhu, Wei Wang, Yulin Zhang, Ming Li, Jiamin Zhang, Lili Ji, Zhiping Zhao, Rui Zhang, Lin Chen

**Affiliations:** Key Lab of Meat Processing of Sichuan Province, Chengdu University, Chengdu 610106, China; 17612808232@163.com (Y.Z.); wangwei8619@163.com (W.W.); zhangyulincdu@163.com (Y.Z.); liming19970107@163.com (M.L.); jasminejj@163.com (J.Z.); lily_jee@126.com (L.J.); zhaozhiping@cdu.edu.cn (Z.Z.); zhangrui@cdu.edu.cn (R.Z.)

**Keywords:** Mianning ham, deep spoilage, high-throughput sequencing, volatile compounds, physicochemical properties

## Abstract

Deep spoilage is a cyclical and costly problem for the meat industry. Mianning ham is a famous dry-cured meat product in Sichuan, China. The aim of this work was to investigate the physicochemical characteristics, sources of odor, and associated microorganisms that cause spoilage of Mianning ham. High-throughput sequencing and solid-phase microextraction–gas-chromatography (SPME-GC-MS) techniques were used to characterize the physicochemical properties, microbial community structure, and volatile compounds of spoiled Mianning ham and to compare it with normal Mianning ham. The results showed that spoiled ham typically had higher moisture content, water activity (a_w_), and pH, and lower salt content. The dominant bacterial phylum detected in deeply spoiled ham was Firmicutes (95.4%). The dominant bacterial genus was *Clostridium_sensu_stricto_2* (92.01%), the dominant fungal phylum was Ascomycota (98.48%), and the dominant fungal genus was *Aspergillus* (84.27%). A total of 57 volatile flavor substances were detected in deeply spoiled ham, including 11 aldehydes, 2 ketones, 6 alcohols, 10 esters, 20 hydrocarbons, 6 acids, and 2 other compounds. Hexanal (279.607 ± 127.265 μg/kg) was the most abundant in deeply spoiled ham, followed by Butanoic acid (266.885 ± 55.439 μg/kg) and Nonanal (165.079 ± 63.923 μg/kg). *Clostridium_sensu_stricto_2* promoted the formation of five main flavor compounds, Heptanal, (E)-2-Octenal, 2-Nonanone, Hexanal, and Nonanal, in deeply spoiled ham by correlation analysis of microbial and volatile flavor substances.

## 1. Introduction

Dry-cured ham is a traditional fermented meat product that has a long history of development. Dry-cured ham is a raw meat product made from whole pork legs with bones, skin, and claw tips that have been cured, washed, air-dried, fermented for a long time, and shaped [1]. The production of dry-cured ham is more common in Mediterranean countries such as Italy, Spain, or Portugal [2]. In addition, some European countries also have a tradition of dry-cured meats, such as Croatian Prsut and Polish Kindziuk or Kumpiak podlaski [3,4]. Several studies have reported sensory attributes and volatile compounds in qualifying dry-cured hams, especially in Spanish, Italian, and French dry-cured hams, and also Slovenian dry-cured ham [5]. Additionally, dry-cured ham has been largely studied for its physicochemical and sensory characteristics depending on different processing technologies [4]. Mianning ham is a famous fermented meat product in Sichuan, China, protected by the National Geographical Indication, made from plump pig hind legs that are repaired, cured, washed and dried, and fermented. The common types of spoilage in dry-cured ham are deep spoilage and vein defects (surface rot). Deep spoilage, which is typical of the winter months, is caused by *Enterobacter agglomerans, Proteus vulgaris, Serratia liquefaciens,* and *Clostridium* spp. and characterized by the smell of sulfur compounds [6]. Some odors, such as potato, benzoic acid, and walnuts, can be found near the femoral head. Deep spoilage is the most important type of deterioration, where the growth of microorganisms causes the breakdown of proteins, resulting in a paste-like texture of the ham and the production of peptides, amino acids, amines, ammonia, sulfides, alcohols, aldehydes, ketones, and organic acids. It gives the ham an unpleasant and unacceptable taste [7,8]. Surface spoilage which represents 40% of the total dry-cured ham defects, is caused by a microorganism that was recently identified as *Marinilactobacillus psychrotolerans* [9].

Mianning ham is produced using natural fermentation and without the use of any preservatives. The quality of the ham relies entirely on the penetration of salt and dehydration. Therefore, during the production and maturation of ham, it is susceptible to microbial contamination, leading to spoilage and deterioration. It has been suggested that this type of spoilage is caused by microflora, especially bacterial contamination, and the corresponding strains have been isolated based on culture methods [10]. It was also found that fungal toxins and metabolites can cause off-flavors at various stages of ham production and processing, with *Penicillium*, *Aspergillus*, and *Ulva* being the most common, negatively affecting the quality of ham and causing it to spoil [11]. Most researchers attribute the cause of spoilage to bacteria [12], while others point to the influence of meat’s own enzymes on spoilage [13].

The microbial community structure and volatile flavor substances associated with spoilage of Mianning ham have not been reported. Therefore, the present work is on the differences in physicochemical indicators, microbial communities, and volatile compounds between different spoilage types of Mianning ham and normal Mianning ham. It uses physicochemical properties and volatile flavor substances to distinguish normal Mianning ham from rotten Mianning ham and to identify the main microorganisms that cause the rottenness of Mianning ham. This is of great importance to the development of the Mianning ham industry.

## 2. Materials and Methods

### 2.1. Sample Preparation and Sampling

The dry-cured ham samples used in this study were obtained from the hind legs of Wujin pigs crossed with Changbai pigs in Liangshan, Sichuan. The crossbred pigs were fed with corn, buckwheat, bean seeds, crushed material, and agricultural straws. After 7 months of feeding, the live weight was about 150 kg and the pigs were randomly slaughtered. Hind legs of pigs with less fat, thin skin, and lean meat were selected for curing and processing, with a weight of 7.5–10 kg per leg. In a cold storage room at a temperature of 4–5 °C and 90% relative humidity, the legs were cured according to the traditional dry-curing method, using a salt amount of 6% of the weight of the legs for 4 coats. The cured pork legs were washed in clean water to remove excess salt from the muscle surface and to reveal the red color of the muscle surface, and then hung in a ventilated place to dry for 4 days. The dried hams were hung in a ventilated room and fermented naturally for 3 months before sampling.

Mianning ham was provided by a processing plant in Mianning County, taking three each of normal ham, surface spoiled ham and deep spoiled ham, all from the same batch of hams produced under the same external environment and process conditions. The selection of spoiled Mianning ham was performed by a quality-control professional with 30 years of experience in Mianning ham processing, who used a thin bamboo stick to pierce the inside of the ham and detect the initial sings of spoilage (off-odor). Normal Mianning ham has a flat muscle cut and the characteristic aroma of ham. The deep spoiled ham and the surface spoiled ham are similar in that they both have an off-odor. The most significant difference between deep spoiled ham and surface spoiled ham is that the muscles near the bones of deep spoiled ham are mushy. Referring to the literature method, samples were collected with a sterile scalpel from approximately the geometric center of the ham. All samples were taken from the biceps femoris muscle and adductor muscles of hams. Next, 3.5 cm–4.0 cm thick meat was cut from the normal hams as normal ham samples (ZC); 3.5 cm–4.0 cm of meat was cut from the surface spoiled hams as superficial spoilage samples (FBQ); and finally, samples were taken from deep spoiled hams at a depth of 3.5 cm–4.0 cm near the bone as deep spoilage samples (FBS) [14]. All of the samples were immediately stored at −80 °C for subsequent analysis.

### 2.2. Physical and Chemical Index Measurements

The pH and water activity (a_w_) were measured according to the methods described by Wang et al. [15]. The pH value was measured using a homogenate prepared with 3 g of sample and distilled water (27 mL), and a pH meter (FE20 Benchtop pH meter, Mettler Toledo Instruments Co., Ltd., Shanghai, China). The a_w_ value was determined by a Fast-lab water-activity meter (Gbx, Romans, France) at 25 °C. The moisture content of the hams was determined after dehydration at 100 °C to constant weight following official methods [16]. The chloride content of the hams was measured using the silver measure method following GB 5009.44-2016 [17]. The malondialdehyde of hams was measured using high-performance liquid chromatography following GB 5009.181-2016 [18]. An HPLC system (Thermo UltiMate 3000 HPLC system, Waltham, MA, USA) with an LPG-3400 SDN pump, a WPS-3000 SL autosampler, a TCC-3000 RS column temperature chamber, and a VMD-3100 UV detector was used. Chromeleon 7 software (Bannockburn, IL, USA) was used. A C_18_ column (250 mm length × 4.6 mm inner diameter) was obtained from Thermo. Mobile phase is 0.01mol/L. The column temperature was 30 °C. The flow rate was 1.0 mL/min. The injection volume was 10 μL and the detection wavelength was 532 nm. The color measurements were measured according to the methods described by Li et al. [19]. Color measurements were determined by a CR-400 portable colorimeter (Konica Minolta Co., Ltd., Tokyo, Japan). The spectro-colorimeter was calibrated against white and black reference tiles covered with same film as that used for ham samples. Lightness (*L**-value), redness (*a**-value), and yellowness (*b**-value) were recorded. All indexes were measured in triplicate.

### 2.3. High-Throughput Sequencing

The main reagents used were MagPure Soil DNA LQ Kit (D6356-02, Magen, Guangzhou, China), Qubit dsDNA Assay Kit (Q32854, Thermo Fisher Scientific, Waltham, MA, USA), and Tks Gflex DNA Polymerase (R060B, Takara, Beijing, China). The main instruments were Tabletop high-speed centrifuge (Centrifuge 5418, Eppendorf, Hamburg, Germany), PCR instrument (580BR10905, Bio-rad, Richmond, CA, USA), QIAxtractor (SN 002358, QIAGEN, Hilden, Germany), Electrophoresis instrument (HE-120, Tanon, Shanghai, China), Gel imager (2500, Tanon, Shanghai, China), Bioanalyzer (2100, Aglient, Santa Clara, CA, USA), and NanoDrop (2000, Thermo Fisher, Waltham, MA, USA).

Genomic DNA was extracted from the samples using a DNA extraction kit, followed by agarose gel electrophoresis and Nanodrop 2000 to detect the concentration of DNA. Genomic DNA was used as template for PCR, using specific primers with barcode and Takara’s Tks Gflex DNA Polymerase according to the selection of a sequencing region to ensure amplification efficiency and accuracy. The V3-V4 region of bacterial 16S rRNA was amplified using generic primers (343F: 5′-TACGGRAGGCAGCAG-3′ and 798R: 5′-AGGGTATCTAATCCT-3′) combined with adapter sequence and barcode sequence, and the fungal ITS region was amplified using primers (ITS1F: 5′-CTTGGTCATTTAGAGGAAGTAA-3′ and ITS2: 5′-GCTGCGTTCTTCATCGATGC-3′) combined with an adapter sequence and a barcode sequence.

The first PCR amplification system was 50 μL, including 10 μL buffer, 0.2 μL Q5 High-Fidelity DNA polymerase, 10 μL high-GC enhancer, 1 μL nucleotides, 10 μmol/L each of forward and reverse primers, and 60 ng of genomic DNA. The amplification procedure was as follows: predenaturation at 94 °C for 4 min, denaturation at 96 °C for 1 min, annealing at 45 °C for 1 min, extension at 72 °C for 1 min, and finally, extension at 74 °C for 6 min. The PCR products were purified using AMPure XP Beads and then quantified using qubit3.0.

The second round of the PCR amplification system was 40 μL, including 20 μL 2 × Phusion HF MM, 8 μL ddH_2_O, 10 μmol/L each of forward and reverse primers, and 10 μL of PCR products amplified in the first round. The amplification procedure was as follows: predenaturation at 94 °C for 4 min; denaturation at 97 °C for 20 s, annealing at 64 °C for 30 s, extension at 72 °C for 1 min, 10 cycles; and finally, extension at 74 °C for 6 min. The PCR products were purified using AMPure XP Beads, and quantitative analysis was performed using an ABI StepOnePlus real-time PCR system (Life Technologies, City, State Abbreviation, Gaithersburg, MD, USA). Finally, at 97% similarity level, the valid data were clustered using QIIME v1.8.0 to obtain operational taxonomic units (OTUs), and the OTUs were annotated with species taxonomic information based on the Silva taxonomic database.

### 2.4. Determination of Flavor Compounds

The extraction method for the flavoring substances was headspace solid-phase microextraction (SPME). Normal and spoiled Mianning ham samples were cut into pieces and precisely 3.00 g was weighed into 15 mL headspace bottles. Then, 1 µL of 2,4,6-trimethylpyridine was added to the headspace bottle as an internal standard, and the headspace bottle was sealed. To adsorb volatile compounds, a SPME fiber (50/30 μmCAR/PDMS/DVB) was extended through the needle and exposed on the headspace of the vial for 30 min at 60 °C [20,21]. The sample pretreatment conditions were set by the CTC autosampler: heating chamber temperature of 75 °C, heating time of 45 min, sample extraction time of 20 min and desorption time of 5 min. Volatiles were analyzed with gas chromatography-mass spectrometry (GC-MS). The sample preparation parameters in the autosampler have been tested before and the most optimal ones were chosen.

Gas chromatography conditions: HP-5ms-UI chromatographic column (30 mm × 0.25 mm, 0.25 mm) was used; the pressure was 32.0 kpa; the column flow rate was 1.0 mL/min; the carrier gas was helium for splitless injection. The injection port temperature was 250 °C.

Column temperature program: the starting temperature is 35 °C for 20 min. It rose to 200 °C at a rate of 5 °C/min, and finally rose to 250 °C at a rate of 15 °C/min and was held for 5 min.

Mass spectrometry conditions: electron ionization source (EI) was used; the electron energy was 70 eV; the temperature of the ion source was 250 °C; the temperature of the transmission line was 150 °C; the mass scan range was 35–500 *m*/*z*; and the scan rate was 1 scan/s. The detector voltage was 350 V.

Qualitative analysis: The chromatograms of the resulting samples were integrated, searched, and compared in the NIST database, and matched with the volatile compounds corresponding to the peaks on the chromatograms, with a match of 80% for the L library.

Quantitative analysis: The relative content of each component was obtained by normalizing the peak area of the total ion flow chromatogram.

### 2.5. Statistical Analysis

Microsoft Excel 2019 (Microsoft, Redmond, WA, USA) was used for data statistics, and IBM SPSS Statistics 22.0 (IBM, Chicago, IL, USA) was used for *t*-test of variance. Cluster heat maps were drawn based on the absolute content of each subject flavor substance with the R (4.1.3) Pheatmap package. Pearson correlation coefficients of subject flavor substances with dominant bacterial and fungal genera were calculated by IBM SPSS Statistics 22.0, and a correlation network map was drawn using Cytoscape (3.9.1) (Bethesda, MD, USA) based on Pearson correlation coefficients of subject flavor substances with dominant bacterial and fungal genera. The results showed that *p* > 0.05 was not significant. However, 0.05 > *p* > 0.01 was significant, and *p* < 0.01 was extremely significant.

## 3. Results and Discussion

### 3.1. Physical and Chemical Index Analyses

Physicochemical characteristics are closely related to the quality of ham, and the physicochemical indices of deeply spoiled ham, surface spoiled ham, and normal ham are shown in Table 1.

The moisture content of deeply spoiled ham (44.7 ± 1.032 g/100 g) was significantly higher than that of surface spoiled ham (39.99 ± 0.146 g/100 g) and normal ham (38.85 ± 0.33 g/100 g) (*p* < 0.05), due to the way Mianning ham is preserved (natural hanging) resulting in a soft interior and a hard exterior, with a higher moisture content inside than on the surface. The higher moisture content may be one of the causes of ham spoilage. The moisture content of spoiled dry-cured Spanish ham was higher than that of normal ham, which is close to the results of this study [22].

Water activity (a_w_) refers to the state of the presence of water in the food and the degree of water binding to the food. Most of the water activity in fresh meat is above 0.98, and various microorganisms grow and multiply vigorously. The a_w_ of deeply spoiled ham was 0.945 ± 0.042, which was significantly higher than that of surface spoiled ham (0.88 ± 0.007) and normal ham (0.852 ± 0.006) (*p* < 0.05). Studies have shown that dry-cured hams are more susceptible to microbial contamination at when a_w_ > 0.91, and it is necessary to reduce aw or keep aw at 0.9 to inhibit the growth of most bacteria [23].

Studies have shown that when pH > 6.0, ham is more susceptible to microbes [24]. The pH values of deeply spoiled and surface spoiled ham were 6.51 ± 0.15 and 6.31 ± 0.01, respectively, which was significantly higher than the normal ham (5.93 ± 0.15) (*p* < 0.05). A higher pH value is one of the indicators of spoilage of meat products. The pH of St. Daniel’s spoiled ham is 6.5 ± 0.3, and the pH of the undenatured ham was 5.9 ± 0.2, which is closer to the present determination of the Mianning ham [25]. Ham will have a better color and texture when the pH is between 5.6 and 6.0 during the processing [26].

The chloride content of deeply spoiled ham was 5.18 ± 0.3 g/100 g, which was significantly lower than that of surface spoiled ham (6.16 ± 0.17 g/100 g) and normal ham (8.29 ± 0.38 g/100 g) (*p* < 0.05). The low chloride content may be related to the salting process of Mianning ham, where salt is applied to the surface of the ham during curing. The surface salt has difficulty penetrating into the interior of the ham. The inside of ham with low salt concentration with high a_w_ very easily leads to deep spoilage. Lower chloride concentrations lead to excessive protein hydrolysis, implying higher tissue enzyme activity. It has been shown that the 5–6% salt content of dry-cured ham completely inhibited the activity of calcium-activating factor and tissue proteinase D in biceps femoris and semimembranosus muscles [27]. The lower salt content may also lead to an increase in pH. The higher the pH, the better the ham’s ability to hold water, which is one of the reasons why the moisture content of spoiled ham is higher than normal ham. A study confirmed that the use of 18% potassium lactate + 12% lysine + 70% NaCl instead of 30% NaCl for fermented dry-cured ham resulted in better quality and flavor, with a 15.71% reduction in salt content [28].

The level of malondialdehyde, the end product of fat oxidation, reflects the degree of spoilage of the ham. The malondialdehyde content of deeply spoiled ham and surface spoiled ham was 1.93 ± 0.15 mg/kg and 1.55 ± 0.13 mg/kg, respectively, which was significantly higher than that of normal ham (0.98 ± 0.94 mg/kg) (*p* < 0.05).

The muscle color of dry-cured ham is one of the most important indicators to evaluate its eating quality. The main chromogenic substances in ham are myoglobin-like and hemoglobin and its derivatives [29]. The *L** (lightness) and the *a** (redness) directly reflect the color quality of meat and meat products and are widely used for meat color analysis. The *L** of the three ham samples were significantly different (FBS > FBQ > ZC) (*p* < 0.05), and the color of spoiled ham was much lighter than that of normal ham. It was found that lightness was related to the thin layer of water on the surface of the muscle tissue, implying that moisture and hydration affect the lightness of dry-cured hams. Therefore, the higher moisture content in the deeply spoiled ham and surface spoiled ham may be a factor for the higher lightness values of these hams [30]. The *a** of surface spoiled ham and normal ham was significantly higher than for deeply spoiled ham (*p* < 0.05). A spectroscopic analysis of pigmented material in Parma ham conducted by Jens et al. showed that large amounts of pigments were produced with increasing curing and fermentation time, but these pigments were not nitroso-myoglobin; they were probably some compounds containing trivalent iron ion ligands with good oxidative stability [31]. This may explain the redder coloration of spoiled ham [32]. It has been shown that the *a** is highly positively correlated with the NaCl concentration [33]. The *b** (yellowness) is usually extremely unstable during processing, but dry-cured ham exhibits relatively stable *b** values due to the long processing period. The *b** of deeply spoiled ham (9.32 ± 0.58) was significantly higher than that of surface spoiled ham (8.25 ± 0.26) and normal ham (7.46 ± 0.22) (*p* < 0.05). The *b** increases when oxygenation or oxidation of myoglobin occurs [34].

### 3.2. Bacterial Diversity Analysis

The bacterial α-diversity indices of normal and spoiled Mianning ham are shown in Table 2. There are nine samples in the project. The data volume of clean tags after quality control is distributed between 71,152–78,043, and the data volume of clean tags after removing chimeras to obtain valid tags (the final data for analysis) is distributed between 60,485–74,897. The average length of valid tags was 402.4–418.22 bp, and the number of OTUs in each sample was distributed between 542–1176. OTU classification of quality sequence valid tags obtained from QC according to 97% similarity was performed using Vsearch (version 2.4.2) software (an open source tool for microbiome analysis) [35]. Based on the results of OTUs clustering analysis and research needs, the number of OTUs shared and unique among different samples was analyzed and plotted as a Venn diagram (Figure 1). The three ham samples had a total of 661 bacterial OTUs.

The number of OUTs was lower in deeply spoiled ham than in surface spoiled ham and normal ham (*p* > 0.05), and the highest number of OUTs was found in normal ham. There was a significant difference (*p* < 0.05) in the bacterial abundance between deeply spoiled ham and surface spoiled ham; there were fewer bacterial species present in deeply spoiled ham than in surface spoiled ham. There was a significant difference (*p* < 0.05) in the bacterial diversity among deeply spoiled ham and surface spoiled ham and normal ham, and the diversity of bacterial communities in deeply spoiled ham was lower than surface spoiled ham and normal ham. The coverage of all three ham samples was 0.999, indicating that the sequencing depth of the method was sufficient to reflect the bacterial community of the samples and the data could be used for subsequent analysis.

To further understand the bacterial community structure of deeply spoiled ham, surface spoiled ham and normal ham, this experiment analyzed the colony composition of Mianning ham at both phylum and genus levels. The bacterial abundance was visualized in the form of a stack diagram. At the phylum level (Figure 2a), the top 30 microbial phyla in terms of abundance, such as Firmicutes, Bacteroidota, Proteobacteria, and Actinobacteriota. The dominant phylum in deeply spoiled ham was Firmicutes, which accounted for 95.4% of the total number of bacteria that dominated the spoilage of ham. In Jinhua ham, Proteobacteria, Bacteroidetes and Firmicutes dominated the bacterial populations, which were closer to the results of this test [36]. Meanwhile, Bacteroidota and Proteobacteria were the dominant phylum in surface spoiled ham and normal ham. At the genus level (Figure 2b), the top 30 microbial genera in terms of abundance, such as *Clostridium_sensu_stricto_2*, *Leptotrichia*, *Prevotella* and *Muribaculaceae*, were identified. *Clostridium_sensu_stricto_2* was the dominant genus, with 92.01% in deeply spoiled ham and 5.38% and 0.03% in surface spoiled and normal ham, respectively. *Clostridium_sensu_stricto_2* may have caused or promoted the spoilage of the ham to some extent. Meanwhile, the dominant genus of surface spoiled ham and normal ham was *Leptotrichia*.

### 3.3. Fungal Diversity Analysis

The fungal α-diversity indices of normal and spoiled Mianning ham are shown in Table 3. A total of nine samples were tested in the project. The data volume of raw reads in the sequencing down machine was distributed between 78,310–81,617, and the data volume of clean tags after quality control was distributed between 14,512–50,512. The valid tags (the data finally used for analysis) were obtained after noise reduction and removal of chimeras using DADA2 in QIIME2 [37]. The data volume is distributed between 14,512–50,436. Each de-duplicated sequence is called ASVs, and the number of ASVs in each sample is distributed between 7 and 40. A Venn diagram (Figure 3) was used to analyze and compare the common and unique ASVs among and within groups to preliminarily understand the ASV characteristics among groups. Three ham samples had a total of seven fungal ASVs.

There was a significant difference (*p* < 0.05) in the number and abundance index of ASVs between normal ham and two different levels of spoiled ham, indicating that more fungal species were present in deeply spoiled ham and surface spoiled ham than in normal ham. There was a significant difference (*p* < 0.05) in the diversity index between deeply spoiled ham and surface spoiled ham and normal ham, indicating that the fungal community in deeply spoiled ham was higher than that in surface spoiled ham and normal ham. The fungal coverage of both spoiled and normal ham samples was 0.999, indicating that the method has basically covered the fungal community diversity and the measured data can be used for subsequent analysis.

To further investigate the spoilage mechanism of Mianning ham and to understand the differences in fungal communities between different levels of spoiled ham and normal ham, this experiment was conducted to statistically analyze the fungal community composition of Mianning ham at both phylum and genus levels. The fungal abundance was visualized in the form of a stack diagram. At the phylum level (Figure 4a), Ascomycota, Basidiomycota and Glomeromycota were identified. Among them, Ascomycota accounted for 98.48%, 72.74%, and 92.81% in deeply spoiled ham, surface spoiled ham, and normal ham, respectively, and was the dominant phylum in all three ham samples. Ascomycota was the dominant phylum in the superficial and internal fungal communities of normal Mianning ham, accounting for 99.72% (superficial) and 97.49% (internal) of the total number of fungi, respectively, followed by Basidiomycota, which is close to the results of the present assay [38]. At the genus level (Figure 4b), only the dominant microorganisms in the top 30 in terms of abundance were shown, with the genera *Aspergillus, Wallemia, Debaryomyces, Yamadazyma* and *Robillarda* detected, with *Aspergillus* accounting for 84.27% of the fungal populations in deeply spoiled ham and normal ham, respectively, and 69.4%. The dominant genera in Panxian ham were *Aspergillus* and *Penicillium*, which was similar to the results of the present test for Mianning ham [39]. The genus with the highest percentage of surface spoiled ham was *Aspergillus* (34.23%), followed by *Wallemia* (25.1%) and *Debaryomyces* (22.21%). It has also been found that *Aspergillus* is the only representative of *Aspergillus* detected from spoiled ham, but this species is not usually found on dry-cured ham. Therefore, *Aspergillus* in surface spoiled ham may be the result of contamination of the ham surface after contact with the environment [40].

### 3.4. Flavor Compound Analysis

The types and contents of volatile compounds in normal and spoiled Mianning ham are shown in Table 4, and a total of 94 volatile flavor substances were detected. A total of 57 flavor substances were detected in deeply spoiled ham, including 11 aldehydes, 2 ketones, 6 alcohols, 9 esters, 20 hydrocarbons, 6 acids and 2 other compounds. Meanwhile, 51 flavor substances were detected in surface spoiled ham, including 7 aldehydes, 6 alcohols, 9 esters, 25 hydrocarbons, and 4 acids. Finally, 42 flavor substances were detected in normal ham, including 8 aldehydes, 3 ketones, 5 alcohols, 3 esters, 19 hydrocarbons, and 4 acids. More volatile compounds were found in deeply spoiled ham than in surface spoiled ham. Hexanal (279.607 ± 127.265 μg/kg) had the highest content in deeply spoiled ham, followed by Butanoic acid (266.885 ± 55.439 μg/kg) and Nonanal (165.079 ± 63.923 μg/kg). As recently shown, 3-methyl-butanal was one of the largest contributors to the flavor of Istrian dry-cured hams [41]. The most abundant aldehyde in the hams was hexanal generated by lipid oxidation. It was particularly high in Sanchuan ham at 6.52%, which revealed that the oxidation extent of Sanchuan ham was higher than Mianning ham [42]. Hexanal content has been widely used to monitor oxidative stability in meat and meat products [43]. Benzeneacetaldehyde (251.58 ± 19.643 μg/kg) had the highest content in surface spoiled ham, followed by butanoic acid (226.173 ± 90.197 μg/kg) and benzaldehyde (251.58 ± 19.643 μg/kg). Phenylacetaldehyde, another Strecker aldehyde, provided hams with a honey-like odor [19]. Normal ham had the least variety in volatile compounds, the highest of which was hexanal (169.272 ± 2.965 μg/kg), followed by nonanal (99.869 ± 14.441 μg/kg) and benzaldehyde (96.138 ± 1.627 μg/kg). Acids were present in a lower amount in all the hams: 8.36% in Mianning ham, 7.15% in Nuodeng ham, 3.18% in Saba ham, and 2.35% in Sanchuan ham. Amongst these compounds, the most abundant acids were 3-methyl-butanoic acid, followed by acetic acid, 2-methyl-propanoic acid, and butanoic acid [41].

To further investigate the contribution of volatile flavor substances to the overall flavor characteristics of Mianning ham when it spoils, the OAV of each substance was calculated based on the absolute content and sensory threshold of each flavor substance. It is usually considered that volatile flavor compounds contribute more to the overall flavor when OAV ≥ 1 [44]. Overall, 10 main flavor substances contributing to the overall flavor characteristics of Mianning ham were screened by OAV, including 7 aldehydes, 1 alcohol, 1 ketone, and 1 ester. The analysis was performed using a cluster heat map (Figure 5). Five of these substances, heptanal, (E)-2-octenal, 2-nonanone, hexanal and nonanal, were present in high levels in deeply spoiled ham and contributed to the flavor of the spoiled ham to the greatest extent, potentially being the source of the slightly acidic or spoiled odor. Methional, benzeneacetaldehyde, and decanoic acid ethyl ester contributed the most to surface spoiled ham. The contribution value of 1-Octen-3-ol to the aroma was greater in normal Mianning ham. However, their abundance was rather low, as indicated by 2-heptanone and 1-octen-3-one with 0.21% and 0.03% in Nuodeng ham, respectively [42].

Aldehydes usually have low thresholds and strong odors and play an important role in meat product flavor formation, where most straight-chain aldehydes are produced by auto-oxidation and oxidative deamination–decarboxylation of unsaturated fatty acids and most branched-chain aldehydes are produced by Strecker degradation of amino acids [4]. Spoiled ham contains higher levels of aldehydes than normal ham, with hexanal and nonanal being the most abundant aldehydes in many dry-cured hams. In mature Kumpiak podlaski ham, hexanal is the predominant flavor compound [45]. Large amounts of hexanal are often found in Iberian and Parma hams, which is considered as distinctive trait of these products [46]. Hexanal has a grassy smell, while heptanal, octanal, and nonanal have a fatty aroma [47]. These aldehydes have a low threshold and generally play a positive role in the formation of product aroma at low concentrations, while at higher concentrations, they become a major source of off-flavors [46]. This is one of the possible reasons for the formation of odor in deeply rotten hams.

Ketones are intermediate products of lipid oxidation and can be formed by the chemical auto-oxidation of free fatty acids [48]. 2-Nonanone has a soapy odor, which may be related to the formation of off-flavors in the late stages of ham storage and contributes to the odor of deeply spoiled ham. 2-Nonanone is formed in small quantities in chemical reactions but in a large part is derived from micro-organism metabolism such as carbohydrate fermentation, lipid and ethyl esters oxidation, and amino acid catabolism [49]. Therefore, the high content of 2-nonanone in our present study indicates the development of spoilage microflora during ripening. García et al. reported similar results, suggesting that the increase in ketones in Iberian ham was related to the activity of spoilage microorganisms [50]. In previous research 2,3-octanedione was detected in Shinkenspeck and Parma ham. 2,3-Octanedione is rarely either present or the dominant ketone in the profile of volatile compounds in dry-cured hams. Spanish and French hams have the highest concentrations of 2-propanone [3].

The production of alcohols is closely related to amino acid metabolism, fat degradation and oxidation, methyl ketone reduction, and microbial growth and reproduction [51]. Usually, alcohols have a high threshold and contribute less to the flavor of the product. However, some unsaturated alcohols were detected in all three hams, such as 1-octen-3-ol and 2-propyl-1-pentanol. These alcohols have low thresholds and may play an important role in aroma variation. Among them, 1-octen-3-ol, formed by the oxidation of arachidonic acid, is frequently detected in dry-cured ham and is considered to be the characteristic flavor substance of Panxian ham, Jinhua ham, and Rugao ham [19,52].

Esters are derived from the esterification of carboxylic acids and alcohols. Esters of short-chain acids have a fruity and sweet taste, but long-chain acids produce a fatty odor [53]. Most esters have a low sensory threshold, and the aroma of dry cured ham is often attributed to the fruity odor of the ester [54]. Thirteen esters were detected in spoiled Mianning ham, while only three esters were detected in normal Mianning ham. Another source of esters may be the unique microbiota that emerge during the ripening period of dry-cured ham. Previous findings indicated that the process of curing Spanish dry-cured ham provides an ecosystem suitable for the survival of staphylococcus and micrococcus from the family *Micrococcaceae* [55]. In the intramuscular tissue halotolerant bacteria of traditional Spanish dry-cured ham, most Gram-positive catalase-positive cocci undergo deamination and decarboxylation processes, and due to microbial esterase activity, esterification with bacteria and mycobacteria on the surface of the ham [56].

Fewer acids were detected in Mianning ham, and free carboxylic acids can be produced from the hydrolysis of triglycerides or phospholipids in ham. The highest absolute content of deeply spoiled ham was butanoic acid (266.885 ± 55.439 μg/kg), followed by 2-methyl butanoic acid (123.936 ± 28.854 μg/kg) and 3-methyl butanoic acid (62.521 ± 16.649 μg/kg). Unlike other studies, 3-methyl butanoic acid was detected in both spoiled and normal Mianning ham, but the level of 3-methyl butanoic acid was lower in normal ham (8.939 ± 1.256 μg/kg). The high content of 3-methyl butanoic acid in spoiled ham can be explained by the oxidation of 3-methylbutyraldehyde to form 3-methylbutyric acid and the high protease activity of Enterobacteriaceae that promotes the release of free amino acids, which are generated by microbial fermentation to produce 3-methylbutyric acid [57]. Butanoic acid and 3-methyl butanoic acid can provide cheese flavor to ham [58]. Octanoic acid and capric acid were detected in all three types of ham, octanoic acid having the taste of sweat and cheese, and capric acid having the taste of sourness. These substances may have an effect on the formation of off-flavors in Mianning ham during the later stages of storage.

A total of 39 alkanes and 4 olefins were detected in Mianning ham. The n-alkanes may be produced by the automatic oxidation of fats, while the branched alkanes may be produced by the oxidation of branched fatty acids in the feedstock. Alkanes are mostly weak or tasteless in aroma, but hydrocarbons may have a fundamental role in flavor formation as important intermediates of heterocyclic compounds. Olefins can be potentially useful for flavor formation as precursor substances for aldehydes and ketones [59].

### 3.5. Analysis of Microbial Association with Key Flavor Substances

The processing of dry-cured ham is accompanied by a series of biochemical and enzymatic reactions, including protein hydrolysis, lipid oxidation, merad reaction, and others. These reactions are important for the formation of ham flavor, and the enzymatic action of microorganisms also contributes to the production of flavor compounds [60]. Huan et al. concluded that microorganisms are the key factors affecting the flavor of different grades of Jinhua ham [54]. Based on the analysis of the dominant microbial genera and key volatile flavor substances of three different quality hams, the correlation network between microbial genera and subject volatile flavor substances was constructed using Pearson correlation coefficients (Figure 6). It was concluded that 15 bacterial genera and 10 fungal genera with 10 key flavor substances form a complete network. *Clostridium_sensu_stricto_2* was the dominant bacterial genus in deeply spoiled ham (Figure 2b), with highly significant positive correlations with heptanal, (E)-2-octenal, 2-nonanone and (E)-2-nonenal (*p* < 0.01), there was a significant positive correlation with hexanal and nonanal (0.05 > *p* > 0.01). *Clostridium_sensu_stricto_2* promoted the formation of five main flavor compounds in deeply spoiled ham and may dominate the spoilage of ham, consistent with the results in Figure 6. *Wallemia*, *Debaryomyces* and *Rhodotorula* had a highly significant positive correlation with benzeneacetaldehyde (*p* < 0.01). Phenylacetaldehyde and benzaldehyde are substances with high content in surface-decayed ham; the formation of phenylacetaldehyde may be promoted by *Wallemia*, *Debaryomyces,* and *Rhodotorula.* Higher levels of benzaldehyde in spoiled ham have been reported [59]. This compound with a bitter almond and acorn odor is frequently found in dry-cured ham and is considered to be the odor active component of Iberian ham [43]. However, the dominant fungal genus *Aspergillus* detected in spoiled ham had no significant positive correlation with any of the 10 key flavor substances. *Ralstonia* had a significant positive correlation with 1-octen-3-ol (0.05 > *p* > 0.01), promoting the formation of 1-octen-3-ol in normal Mianing ham.

## 4. Conclusions

Spoiled ham usually has physicochemical characteristics such as high moisture content, high a_w_, high pH, and low salt content, which make it more susceptible to microbial contamination. Lightness is associated with a thin water-layer on the surface of muscle tissue. The higher moisture content in the deeply spoiled ham and surface spoiled ham may be a factor for the higher lightness values of these hams. Redness made up by nitrosylmyoglobin (MbFe(II)NO) is one of the most important color parameters for cured-meat products such as dry-cured ham, whereas the high lightness level is undesirable. There is evidence that deep spoilage of Mianning ham is caused by the abnormal growth of the Gram-positive bacteria Firmicute, *Clostridium_sensu_stricto_2* and the putrefactive *Ascomycota*. A total of 95 volatile flavor substances were detected by GC-MS. High levels of aldehydes and ketones were detected in spoiled Mianning ham. Hexanal was highest in deeply spoiled Mianning ham, but could not be used to distinguish spoiled ham from normal ham because hexanal was also detected in normal Mianning ham. Compared to other dry-cured hams, large amounts of hexanal are often found in Iberian and Parma hams, which is considered a distinctive trait of these products. The spoiled Mianning ham is saturated with aldehydes and ketones like Spanish ham or prosciutto. 2-Nonanone is largely extracted from the metabolism of microorganisms and its high content in hams indicates the development of spoilage microorganisms during maturation. Therefore, in this study, 2-nonanone can be used to distinguish spoiled Mianning ham from normal Mianning ham during the ripening period. The content of esters in spoiled Mianning ham is much greater than that in normal Mianning ham. Previous findings indicated that the curing process of Spanish dry-cured ham provides an ecosystem suitable for the survival of staphylococcus and micrococcus species of the *Micrococcaceae* family, which undergo esterification with bacteria and molds on the surface of the ham due to the esterase activity of the microorganisms. Therefore, the excess of ester species in Mianning ham can be considered ham with spoilage characteristics. Additionally, 1-chloropentane and 2-methyl butanoic acid were only detected at high levels in deeply spoiled ham, indicating that these compounds were the key discrimination between normal and defective Mianning hams. This study reveals a correlation between the microbial genera of Mianning ham and key flavor substances for the first time. *Clostridium_sensu_stricto_2*, the dominant genus in deeply spoiled hams, showed a strong positive correlation with the main flavor substance in deeply spoiled hams, and it is therefore presumed that *Clostridium_sensu_stricto_2* dominates the spoilage of hams.

However, current high-throughput sequencing based on 16S rRNA and ITS cannot accurately identify the microorganisms responsible for the spoilage of Mianning ham. Therefore, the microorganisms should be further characterized using macrogenome sequencing in subsequent studies, and the association between microorganisms and volatile flavor substances should be further validated using proteomics and metabolomics.

## Figures and Tables

**Figure 1 foods-11-01713-f001:**
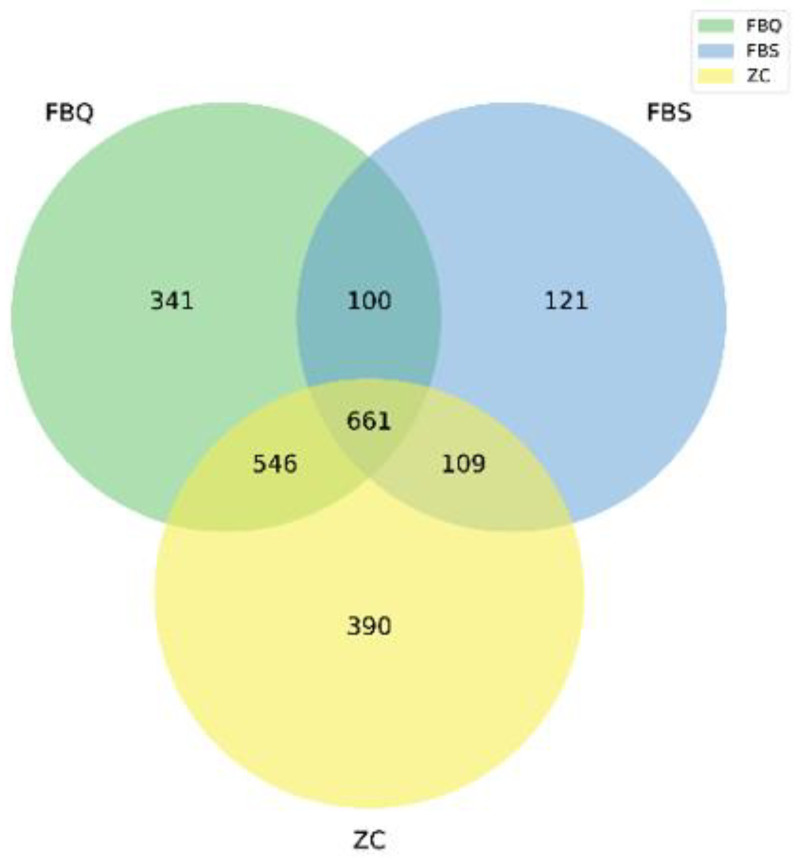
Venn diagram of bacterial diversity of Mianning hams. The figures in different compartments mean the numbers of bacterial specific for or common to Mianning hams of different qualities.

**Figure 2 foods-11-01713-f002:**
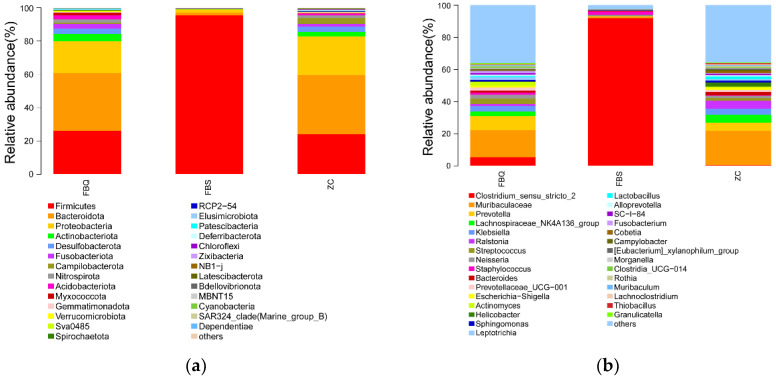
Relative abundance of bacterial community proportions at the phylum (**a**) and genus (**b**) levels in Mianning ham. Different colors represent different bacterial communities, and the size of the color area represents the relative abundance (percentage) of bacterial community at the phylum (**a**) and genus (**b**).

**Figure 3 foods-11-01713-f003:**
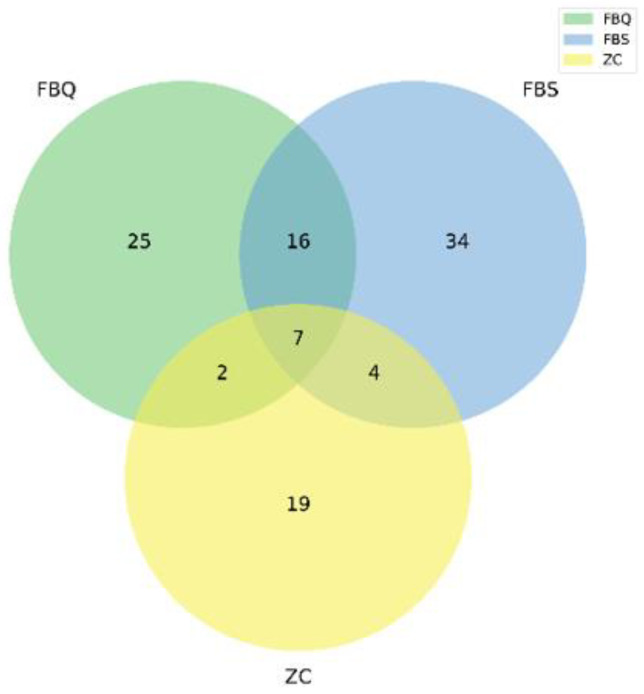
Venn diagram of fungal diversity of Mianning hams. The figures in different compartments mean the numbers of fungal specific for or common to Mianning ham of different qualities.

**Figure 4 foods-11-01713-f004:**
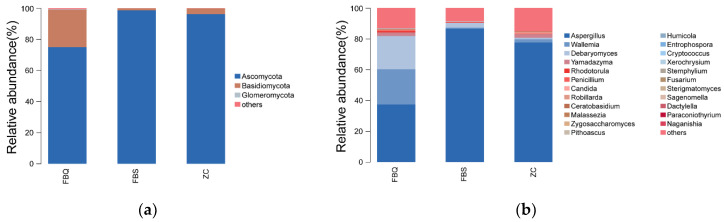
Relative abundance of fungal community proportions at the phylum (**a**) and genus (**b**) levels in Mianning ham. Different colors represent different fungal community and the size of the color area represents the relative abundance (percentage) of fungal community at the phylum (**a**) and genus (**b**).

**Figure 5 foods-11-01713-f005:**
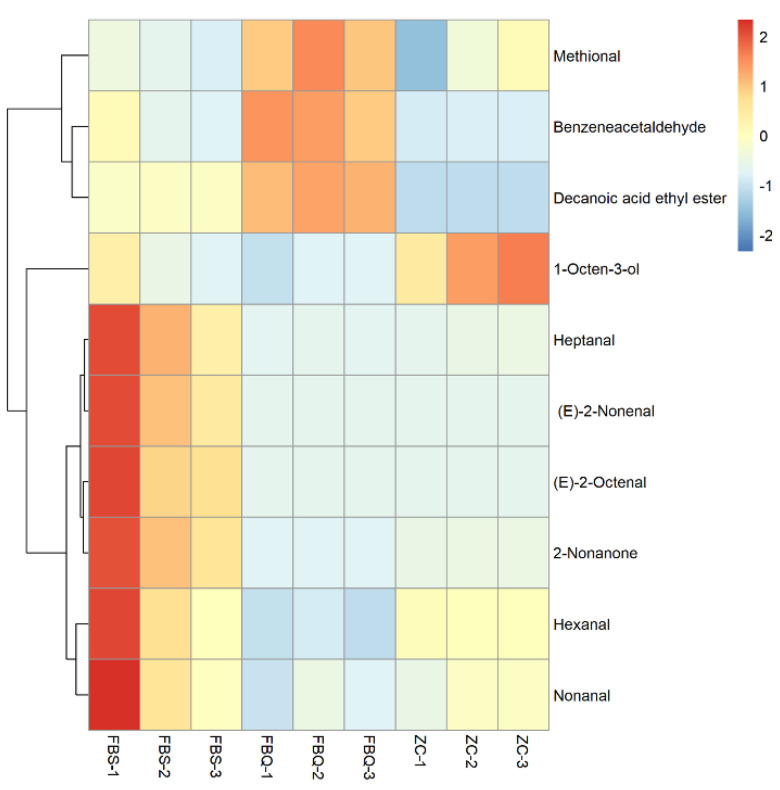
Heat map of hierarchical clustering analysis of main flavor substances in Mianning ham. Each row represents a flavor substance and each column represents a ham sample. The redder the color, the higher the content of volatile compounds; the bluer the color, the lower the content of volatile compounds.

**Figure 6 foods-11-01713-f006:**
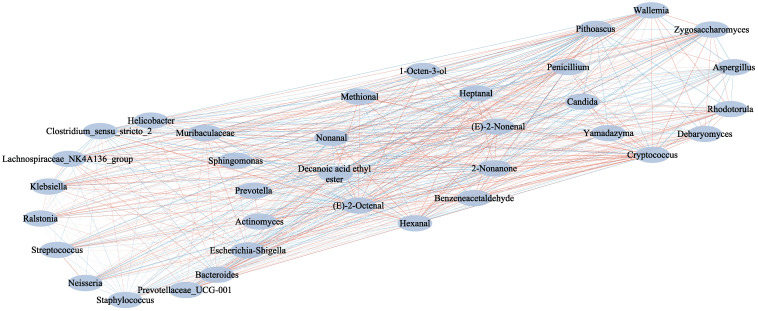
Correlation network of dominant microbial genera and key flavor substances of Mianning ham. The dominant bacterial genus and the dominant fungal genus are on the left and right, respectively. Additionally, the main flavor substance of Mianning ham is in the middle. The red line indicates a positive correlation between the two components and the blue line indicates a negative correlation between the two components.

**Table 1 foods-11-01713-t001:** Physicochemical characteristics of normal and spoiled Mianning hams.

Physicochemical Indexes	FBS	FBQ	ZC
Moisture content (g/100 g)	44.7 ± 1.032 ^a^	39.99 ± 0.146 ^b^	38.85 ± 0.33 ^b^
a_w_	0.945 ± 0.042 ^a^	0.88 ± 0.007 ^b^	0.852 ± 0.006 ^b^
pH	6.51 ± 0.15 ^a^	6.31 ± 0.01 ^b^	5.93 ± 0.15 ^c^
Chloride (g/100 g)	5.18 ± 0.3 ^a^	6.16 ± 0.17 ^b^	8.29 ± 0.38 ^c^
Malondialdehyde (mg/kg)	1.93 ± 0.15^a^	1.55 ± 0.13 ^b^	0.98 ± 0.94 ^c^
Color deviation			
*L**	47.51 ± 0.82 ^a^	44.9 ± 0.22 ^b^	41.76 ± 0.97 ^c^
*a**	10.04 ± 0.86 ^a^	12.53 ± 0.37 ^b^	13.18 ± 0.17 ^b^
*b**	9.32 ± 0.58 ^a^	8.25 ± 0.26 ^b^	7.46 ± 0.22 ^c^

FBS is the deeply spoiled samples of Mianning ham; FBQ is the surface spoiled samples of Mianning ham; ZC is the normal Mianning ham samples. Variance *t*-test: *p* > 0.05 was not non-significant, 0.05 > *p* > 0.01 was significant; *p* < 0.01 was extremely significant. Different letters between groups indicate significant differences (the same below). Means in the same row with different superscripts (a, b, c) are significantly different (*p* < 0.05).

**Table 2 foods-11-01713-t002:** Bacterial alpha diversity index of normal and spoiled Mianning ham.

Samples	Valid Tags	OUT Counts	Observed Species	Chao1	Shannon	Simpson	Good’s Coverage
FBS	74237 ± 585 ^a^	572 ± 40 ^a^	532.07 ± 37.76 ^a^	746.43 ± 61.45 ^a^	0.94 ± 0.65 ^a^	0.153 ± 0.122 ^a^	0.999
FBQ	66418 ± 2492 ^b^	918 ± 219 ^a^	894.6 ± 220.75 ^ab^	1121.15 ± 179.79 ^b^	7.25 ± 0.31 ^b^	0.983 ± 0.004 ^b^	0.999
ZC	64536 ± 4107 ^b^	973 ± 239 ^a^	960.3 ± 235.73 ^b^	1111.26 ± 238.53 ^ab^	7.42 ± 0.46 ^b^	0.985 ± 0.006 ^b^	0.999

Variance *t*-test: *p* > 0.05 was not non-significant, 0.05 > *p* > 0.01 was significant; *p* < 0.01 was extremely significant. Different letters between groups indicate significant differences. Means in the same column with different superscripts (a, b) are significantly different (*p* < 0.05).

**Table 3 foods-11-01713-t003:** Fungal alpha diversity index of normal and spoiled Mianning ham.

Samples	Reads	ASV Counts	Observed Species	Chao1	Shannon	Simpson	Good’s Coverage
FBS	79550 ± 1151 ^a^	33 ± 8 ^a^	32.43 ± 7.84 ^a^	32.64 ± 8 ^a^	1.09 ± 0.31 ^a^	0.293 ± 0.123 ^a^	0.999
FBQ	80481 ± 1600 ^a^	25 ± 3 ^a^	25.23 ± 2.93 ^a^	25.33 ± 3.06 ^a^	2.74 ± 0.23 ^b^	0.772 ± 0.046 ^b^	0.999
ZC	80728 ± 771 ^a^	12 ± 5 ^b^	12.07 ± 5 ^b^	12.07 ± 5 ^b^	1.69 ± 0.68 ^ab^	0.464 ± 0.297 ^ab^	0.999

Variance *t*-test: *p* > 0.05 was not non-significant, 0.05 > *p* > 0.01 was significant; *p* < 0.01 was extremely significant. Different letters between groups indicate significant differences. Means in the same column with different superscripts (a, b) are significantly different (*p* < 0.05).

**Table 4 foods-11-01713-t004:** Types and contents of flavor compounds in Mianning ham.

Number	RT	Compound Name	CAS	Absolute Content (μg/kg)		
				FBS	FBQ	ZC
Aldehydes						
1	4.999	Hexanal	66-25-1	279.607 ± 127.265	45.845 ± 13.239	169.272 ± 2.965
2	10.594	Heptanal	111-71-7	146.683 ± 60.83	16.167 ± 1.886	23.977 ± 5.092
3	11.014	Methional	3268-49-3	21.133 ± 1.372	34.471 ± 2.405	21.443 ± 6.246
4	16.376	Benzaldehyde	100-52-7	95.57 ± 0.38	137.969 ± 25.191	96.138 ± 1.627
5	23.802	Benzeneacetaldehyde	122-78-1	124.554 ± 35.024	251.58 ± 19.643	66.027 ± 16.365
6	25.084	(E)-2-Octenal	2548-87-0	24.828 ± 10.418	-	-
7	28.103	Nonanal	124-19-6	165.079 ± 63.923	73.304 ± 13.831	99.869 ± 14.441
8	31.274	(E)-2-Nonenal	18829-56-6	56.445 ± 24.873	-	-
9	33.769	Decanal	112-31-2	10.004 ± 2.649	-	-
10	36.52	(E)-2-Decenal	3913-81-3	11.632 ± 3.516	-	9.229 ± 2.221
11	58.302	Pentadecanal	2765-11-9	12.053 ± 4.541	-	-
12	58.307	Hexadecanal	629-80-1	-	47.541 ± 0.293	35.492 ± 6.473
Ketones						
13	9.836	2-Heptanone	110-43-0	28.769 ± 5.8	-	7.013 ± 0.703
14	9.848	5-Methyl-2-hexanone	110-12-3	0	-	11.118 ± 1.198
15	27.433	2-Nonanone	821-55-6	60.126 ± 21.97	-	6.974 ± 0.652
Alcohol						
16	8.747	1-Hexanol	111-27-3	-	-	8.113 ± 0.51
17	18.451	1-Heptanol	111-70-6	16.944 ± 6.027	-	-
18	19.052	1-Octen-3-ol	3391-86-4	48.982 ± 18.461	31.173 ± 6.003	95.706 ± 19.483
19	23.341	2-ethyl-1-Hexanol	104-76-7	-	16.212 ± 1.925	-
20	23.341	4-ethyl-Octyn-3-ol	5877-42-9	18.561 ± 6.999	-	9.933 ± 0.341
21	23.347	1-pentanol	58175-57-8	10.354 ± 0.836	18.001 ± 2.605	15.836 ± 0.934
22	26.098	2-butyl-1-Octanol	3913-02-8	-	8.523 ± 1.427	-
23	26.104	(E)- 2-Decenal -1-ol	18409-17-1	-	-	7.025 ± 0.33
24	26.11	trans-2-Undecen-1-ol	75039-84-8	12.257 ± 1.348	-	-
25	28.68	2-butyl-1-Octanol	3913-02-8	7.266 ± 1.7	27.012 ± 0.99	-
26	31.268	2-ethyl-1-Decanol	21078-65-9	-	14.638 ± 2.18	-
Ester						
27	16.423	1-(benzoyloxy)-2,5-Pyrrolidinedione	23405-15-4	165.377 ± 3.105	-	83.853 ± 2.164
28	22.444	3,7-dimethyl-, formate-1,6-Octadien-3-ol	115-99-1	-	7.481 ± 1.123	-
29	24.466	Butyl isovalerate	109-19-3	8.772 ± 1.387	7.813 ± 0.647	-
30	25.061	but-2-yn-1-yl Carbonic acid nonyl ester	1000383-20-5	-	-	13.514 ± 0.886
31	25.125	Carbonic acid nonyl vinyl ester	1000383-25-6	-	11.018 ± 0.607	-
32	25.3	dihydro-4,4-dimethyl-2(3H)-Furanone,	13861-97-7	13.636 ± 2.058	8.062 ± 0.896	-
33	26.285	Octyl chloroformate	7452-59-7	12.364 ± 1.955	-	-
34	26.296	trichloroacetic acid nonyl ester	65611-32-7	27.811 ± 11.905	-	-
35	33.139	Hexanoic acid butyl ester	626-82-4	-	13.763 ± 3.279	-
36	33.139	Hexanoic acid hexyl ester	6378-65-0	16.151 ± 2.407	-	-
37	41.154	dihydro-5-pentyl-2(3H)-Furanone	104-61-0	-	15.36 ± 0.714	-
38	41.154	Sulfurous acid dodecyl hexyl ester	1000309-13-4	14.005 ± 6.619	28.157 ± 2.309	-
39	41.165	Sulfurous acid hexyl undecyl ester	1000309-13-3	-	-	8.525 ± 1.204
40	42.349	Octanoic acid octyl ester	2306-88-9	14.642 ± 6.373	-	-
41	42.354	Butyl caprylate	589-75-3	-	7.843 ± 1.042	-
42	42.739	Decanoic acid ethyl ester	110-38-3	21.881 ± 0.557	49.82 ± 2.207	-
Hydrocarbon						
43	4.218	1-chloropentane	543-59-9	25.442 ± 6.271	-	-
44	17.711	3-methylnonane	5911-04-6	-	8.78 ± 1.937	-
45	22.45	D-Limonene	5989-27-5	-	7.151 ± 0.585	-
46	22.84	(Z)-3-ethyl-2-methyl-1,3-Hexadiene	74752-97-9	8.701 ± 0.808	-	-
47	24.938	2,4-dimethylhexane,	589-43-5	-	10.645 ± 1.818	-
48	26.425	(Z)-5-Tridecene	25524-42-9	-	-	15.73 ± 1.38
49	26.903	2-methyl-10-Undecen-1-al	1000151-82-1	7.987 ± 1.179	-	-
50	27.561	6-methyltridecane,	13287-21-3	8.642 ± 1.601	18.647 ± 1.711	-
51	27.894	2,6-Dimethylnonane,	17302-23-7	13.654 ± 1.61	20.354 ± 0.779	15.315 ± 3.717
52	28.593	Dodecane	112-40-3	-	16.812 ± 6.018	-
53	29.252	3,5-dimethyloctane,	15869-93-9	-	41.43 ± 0.49	-
54	29.351	3,7-dimethyldecane,	17312-54-8	-	8.943 ± 0.205	-
55	29.362	2,6,10-trimethyldodecane,	3891-98-3	8.398 ± 1.251	-	-
56	29.432	4-methyl-5-propylnonane,	62185-55-1		7.466 ± 0.717	-
57	30.412	2,3-dimethylundecane	17312-77-5	9.532 ± 0.9	8.985 ± 1.709	-
58	30.691	5-propyldecane	17312-62-8	13.326 ± 1.066	11.024 ± 2.17	-
59	31.047	5-methylundecane	1632-70-8	20.167 ± 2.22	16.785 ± 3.022	-
60	31.262	4,8-dimethyl-1-Nonanol	33933-80-1	-	14.04 ± 1.569	-
61	31.484	2-methylundecane,	7045-71-8	-	-	11.239 ± 3.49
62	31.828	3-methylundecane	1002-43-3	36.408 ± 23.381	53.402 ± 9.45	7.46 ± 0.401
63	32.143	3,5-dimethyloctane	15869-93-9	-	-	6.604 ± 0.318
64	32.661	I-3-Methyl-5-undecene	74630-67-4	21.155 ± 3.753	16.169 ± 3.763	-
65	32.988	1-Dodecene	112-41-4	11.547 ± 2.086	13.318 ± 1.873	-
66	33.425	Dodecane	112-40-3	82.188 ± 45.032	121.588 ± 31.205	22.564 ± 5.601
67	33.664	Hexadecane	544-76-3	-	-	8.381 ± 0.669
68	33.967	2,4-Dimethyl-undecane	17312-80-0	-	-	2.508 ± 4.344
69	34.148	6-methyldodecane	6044-71-9	-	-	7.347 ± 12.726
70	34.148	2,5-dimethylundecane	17301-22-3	-	-	12.346 ± 12.322
71	34.154	2,2′-(Butane-1,4-diyl)bisoxirane	2426-07-5	9.477 ± 2.642	-	-
72	34.544	4-methyldodecane	6117-97-1	-	-	15.106 ± 5.038
73	35.64	3-Methyl-5-propylnonane	31081-18-2	-	-	3.319 ± 5.749
74	36.176	4,6-dimethyldodecane	61141-72-8	-	-	9.475 ± 2.538
75	36.602	4-ethylundecane	17312-59-3	-	-	21.216 ± 6.139
76	36.602	Tetradecane	629-59-4	-	-	25.601 ± 0.701
77	37.773	2,6,10-trimethyldodecane	3891-98-3	-	-	7.265 ± 0.894
78	38.315	Tridecane	629-50-5	13.195 ± 3.471	11.649 ± 2.811	10.069 ± 2.712
79	38.315	2,6,11-trimethyldodecane	31295-56-4	-	7.062 ± 1.274	9.786 ± 2.34
80	38.321	3-Methyl-5-propylnonane	31081-18-2	-	9.03 ± 1.52	-
81	40.11	7-Methylheptadecane	20959-33-5	10.765 ± 3.115	9.399 ± 0.244	-
82	41.451	3-methyltridecane,	6418-41-3	13.815 ± 8.025	22.399 ± 7.665	-
83	42.745	Tetradecane	629-59-4	44.302 ± 2.749	55.442 ± 18.935	-
84	46.872	Pentadecane	629-62-9	11.66 ± 3.477	11.556 ± 3.388	11.346 ± 2.207
85	50.748	Hexadecane	544-76-3	7.184 ± 0.668	9.361 ± 0.78	-
Acids						
86	5.68	Butanoic acid	107-92-6	266.885 ± 55.439	226.173 ± 90.197	-
87	8.274	Isovaleric acid	503-74-2	62.521 ± 16.649	76.017 ± 0.636	8.939 ± 1.256
88	8.84	2-methyl butanoic acid	116-53-0	123.936 ± 28.854	-	-
89	25.061	Dichloroacetic acid nonyl ester	83004-99-3	-	-	8.792 ± 0.855
90	32.661	4-methyl-1-acetate-1-Hexanol	91367-59-8	7.596 ± 1.563	-	-
91	32.854	Octanoic acid	124-07-2	11.25 ± 1.392	15.199 ± 2.66	20.041 ± 11.028
92	41.725	n-Decanoic acid	334-48-5	8.233 ± 0.986	6.412 ± 1.111	13.123 ± 0.637
other						
93	11.807	4,6-dimethylpyrimidine,	1558-17-4	11.571 ± 1.626	-	-
94	24.927	tetrahydro-5-methyl-trans-2-Furanmethanol	54774-28-6	7.461 ± 2.111	-	-

The symbol “–” denotes that the substance was not detected.

## Data Availability

The data presented in this study are available on request from the corresponding author.

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
