# Peer review of "Characterization of Quality Properties in Spoiled Mianning Ham"

_foods, 2022, doi:10.3390/foods11121713_

Round 1
Reviewer 1 Report
The manuscript is of interest, but contains some drawbacks, which must be revised.
critics:
No clear take home message is given. This must be improved by the authors.
What was the aim of the study. This is as well not clear and must be improved by the authors for a better understanding.
How did you exactly determine between FBS and FBQ. Please Insert a schematic drawing to illustrate the cutting procedure.
Please insert More Information about the aim of the study. This is currently not clear.
Why did you use only one primer pair for detection of 16s RNA. To my knowledge different bacteria species does have different 16s RNA sequentces. Therefore it is possible to distinguish different bacteria. Unclear, must be modified for a better understanding.
It is unclear, what is meant by „a w„ ?? This must be explained in more details within the manuscript („The a w of deeply spoiled ham……).
All figure legends must be improved for a better understanding. Not given so far.
Reviewer 2 Report
The Introduction does not cite important basic publications on the agents and symptoms of spoilage of dry raw hams, and does not sufficently address the practical relevance and preventive measures. Likewise, the conclusios are merely descriptive and lack a clear message of relevance to meat processing.
Reviewer 3 Report
Introduction
line 33-34 – The authors should list some other known dry-cured hams of other nations. Here my recommendations: https://doi.org/10.1016/j.meatsci.2017.12.001, https://doi.org/10.1111/ijfs.14697, https://doi.org/10.1016/j.meatsci.2020.108349
line 37-38- This is a scientific paper, so please do not use colloquial names with quotation marks. Please correct your sentences for language and punctuation. Sentences seem to have been cut off. Please correct them.
Materials and Methods
2.1. Sample Preparation and Sampling – here or in introduction authors must provide information about: breeds of pigs, what weight and age they were when slaughtered, and with what they were fed. Furthermore, please describe the procedure for curing hams starting with the preparation of the meat using salt and spices/herbs, where and under what parameters they were cured and how long, until they were submitted for analysis. All this information is necessary and of great importance for the results obtained, especially the analysis of volatile compounds, and shows the uniqueness of this dry-cured ham.
line 65- Whether samples from all hams were collected from a similar muscle and depth? Was the subcutaneous fat layer discarded?
2.2. Physical and Chemical Index Measurements – add a brief description of all 6 methodologies mentioned in this subsection. Stating the methodology in this way is not sufficient.
2.3. High-throughput sequencing – add names and producers of instruments/devices used in this procedure
2.4. Determination of flavor compounds – give details about kind and parameters of SPME fiber used for extraction of volatile substances. Have the sample preparation parameters in the autosampler been tested before and have the most optimal ones been chosen? - will the authors point to a publication where this procedure has been used before?
line 110- instead of analysis time I think you mean desorption time? that’s correct?
2.5. Statistical Analysis – you done also correlation network and heat maps; please mention this in this subsection - was a different program used for this than the ones listed?
Results and Discussion
Figure 2 is completely unreadable, the font is too small, thus contributing nothing to the article. The graph and accompanying legend should be corrected/enlarged.
Figure 4 also needs a font size correction as it is difficult to read the types of microorganisms.
3.4. Flavor Compound Analysis – lines 295-307- there is lack of discussion/comparison of volatiles profile (groups of volatiles or specificone) of Mianning ham with others dry-cured hams from literature. Here I provide some recommendations: https://doi.org/10.1111/ijfs.14697, https://doi.org/10.1016/j.foodres.2022.110977, https://doi.org/10.1016/j.foodres.2021.110222 This literature items will also be useful to expand the discussion of the results from lines 332-361.
Conclusions
line 414-419 - The authors should remove repetitive information already given. Text in this lines is not a conclusions.
line 421-422- Authors wrote: “The color of spoiled ham is more distinct than that of normal ham.” What did the authors mean by this? Please rewrite this sentence in more scientific way.
line 437- please don’t use “etc.” in scientific paper
References
The authors use a lot of very old literature, some of them published more than 20 years ago. I recommend reviewing the literature used and replacing with newer items from the last 5 -10 years ago, for example those recommended in the comments above.
Round 2
Reviewer 1 Report
The revised version is now improved. All recommendations and suggestions of the reviewer are fulfilled.
Reviewer 2 Report
Line 73/74: The sentence "It can detect spoilage in the early stage of fermentation of Mianning ham" is a statement rather than a research question, and should be reworded into a research question. I do not see clear evidence from the data provided to support this statement. Likewise, the Conclusions should answer or at least comment on the question whether or not the approach is suitable to early detection of spoilage. In their present form, the Conclusions are more like a summary, and do not address sufficiently the aims of the work as outlined in the Introduction.
References should be checked for correct spelling and format (this applies, in particular, to ref nos. 3 (title of Journal), 24 and 31 (authors' given names mistaken as family names), 29 and 57 (format), 10, 14, 42, and 49.
Reviewer 3 Report
2.1. Sample Preparation and Sampling.
Please explain what does it mean, that pigs were Free-feeding ? Give the main ingredients of their feed. Did you use herbs or spices for curing hams?
“The Determination of malondialdehyde of hams was measured using high performance liquid chromatography following GB 5009.181-2016.” – please describe in detail the chromatographic analysis of this compound - give the parameters, column, detector and other important data
Please write in text that: The sample preparation parameters in the autosampler has been tested before and the most optimal ones have been chosen.
3.4. Flavor Compound Analysis
The authors have expanded the discussion of results of volatile profile slightly but not enough. The authors were limited to only comparing the volatiles compound profile of dry-cured ham with domestic dry-cured hams and one foreign. The authors should refer to other European hams. The authors missed the fact that most esters are derived from microbial metabolism as was proven by Karpiński et al. (https://doi.org/10.1111/ijfs.14697) and demonstrated by Table 4 in this article. Please mention that in discussion.
References
Authors should correct the names of the authors of the cited literature. In many places, the names are either messed up or with missing letters. For unusual letters, maybe use symbols from other languages from the Word menu as letters.
